# Ultra-High Frequency Surface Acoustic Wave Sensors for Temperature Detection

**DOI:** 10.3390/mi15010135

**Published:** 2024-01-15

**Authors:** Qi Dong, Qutong Yang, Xiaoyang Liu, Shenghe Hu, Wenzhe Nie, Zhao Jiang, Xiaoming Fan, Jingting Luo, Ran Tao, Chen Fu

**Affiliations:** Key Laboratory of Optoelectronic Devices and Systems of Ministry of Education and Guangdong Province, College of Physics and Optoelectronic Engineering, Shenzhen University, Shenzhen 518060, China; 2110456080@email.szu.edu.cn (Q.D.); jingxicaoshilang2@gmail.com (Q.Y.); 2110456113@email.szu.edu.cn (X.L.); 2210452078@email.szu.edu.cn (S.H.); 2210452084@email.szu.edu.cn (W.N.); 2210452044@email.szu.edu.cn (Z.J.); abyssscale@163.com (X.F.); ran.tao@szu.edu.cn (R.T.)

**Keywords:** surface acoustic wave, wireless, high frequency, temperature

## Abstract

Highly sensitive surface acoustic wave (SAW) sensors have recently been recognized as a promising tool for various industrial and medical applications. However, existing SAW sensors generally suffer from a complex design, large size, and poor robustness. In this paper, we develop a simple and stable delay line ultra-high frequency (UHF) SAW sensor for highly sensitive detection of temperature. A Z-shaped delay line is specially designed on the piezoelectric substrate to improve the sensitivity and reduce the substrate size. Herein, the optimum design parameters of extremely short-pitch interdigital transducers (IDTs) are given by numerical simulations. The extremely short pitch gives the SAW sensor ultra-high operating frequency and consequently ultra-high sensitivity. Several experiments are conducted to demonstrate that the sensitivity of the Z-shaped SAW delay line sensor can reach up to 116.685°/°C for temperature detection. The results show that the sensor is an attractive alternative to current SAW sensing platforms in many applications.

## 1. Introduction

Currently, many electrical and optical temperature sensor components have been developed for ambient temperature detection, e.g., thermocouple sensors [1], thermistor sensors [2], jet temperature sensors [3], infrared temperature sensors [4], and so on. These detection techniques’ inherent optical and electrical properties limit their use in industrial production and aerospace applications. For example, conventional sensors are easily damaged in high-pressure environments, and circuit boards are susceptible to moisture in high-humidity environments, leading to short circuits and other problems. In addition to their poor robustness, conventional sensors have been unable to meet the demands of more demanding applications in terms of sensitivity, response time, measurement limits, and other performance parameters [5,6].

SAW is generated from the free surface of the elastomer and propagates various types of waveforms on the surface; its energy is mainly distributed perpendicular to the surface of the substrate in a region of one to two wavelengths. SAW sensors are mainly fabricated by micro electromechanical system (MEMS) technology using piezoelectric substrates, and its fabrication compared to other types of MEMS devices, has the advantages of simple procedures, stable parameters, and strong repeatability. Currently, SAW sensors have been used in various fields such as mechanical [7], chemical [8], electrical [9] and biological [10]. The SAW sensors consist of a simple structure of mainly piezoelectric materials and surface deposition of metal electrodes. SAW sensors are able to monitor subtle changes on the surface and provide sensitive feedback, resulting in high sensitivity, good reliability, and fast response time. In fact, since piezoelectric materials generally have a good temperature coefficient, they are well suited for potential temperature sensing applications. Upon accomplishment of further structural design, peripheral circuitry, and antenna matching of the SAW sensor, they can also be placed in harsh environments for wireless passive detection [11].

Under the rapid development of the Internet of Things and 5G communication technology, the working frequency and response bandwidth of SAW sensors gradually put forward more requirements in the direction of high frequency and broadband [12]. The sensitivity of SAW sensors is directly related to the working frequency, and the higher the working frequency of SAW sensors, the higher their measurement accuracy. Amongst them, the design of the reflective delay line SAW transducer has been shown to enable high-frequency band operation and easier access to a larger effective delay bandwidth product [13]. In addition, delay line sensors with reflective grids have further reduced size compared to other sensors of the same type. In recent years, several groups have reported some SAW delay line temperature sensors based on reflective gratings [14,15,16]. However, the sensitivity of delay line sensors depends on the delay time. Therefore, conventional parallel delay lines require longer chips in the horizontal direction to obtain higher sensitivity, which hinders the miniaturization of the sensing device and reduces the signal-to-noise ratio [17].

In this paper, two designs of a parallel delay line and a novel Z-path delay line are proposed to investigate the effects of different delay line layouts on temperature sensitivity and miniaturization of the device. The 128° YX Lithium Niobate is chosen as the piezoelectric substrate for UHF SAW delay line sensors with a center frequency of 2.45 GHz. The rest of the paper is organized as follows. First, the working principle and fabrication procedures of the delay line sensor are described. Then, the necessary electrode design parameters are obtained, and the center frequency is estimated using numerical simulation. Then, suitable sensors are screened by performance tests, and finally, temperature sensing experiments are performed to analyze the advantages and disadvantages of the two different delay line structural designs.

## 2. Experiment and Methods

### 2.1. SAW Delay Line Sensor Fabrication

The SAW delay line sensor mainly comprises a single IDT and a certain number of reflective gratings (or reflectors), as shown in Figure 1a. The working principle of the single-port SAW delay line sensor is that the IDT first converts an input electrical signal into a SAW signal. Afterward, the surface acoustic wave propagates through the reflective gratings on the surface of the substrate and is reflected back to the IDT, where the SAW signal is then converted back to an electrical signal as an output, achieving a certain time delay of the electrical signal [18].

In this work, we designed a UHF SAW delay line sensor that operates at gigahertz frequencies using a 128° YX lithium niobate (LN) substrate with a thickness of 0.35 mm. The resonant frequency f of the sensor is defined by
(1)$$f=\nu_{s}/\lambda ,$$
where $\nu_{s}=$ 3965 m/s is the velocity of the Rayleigh wave traveling along the free surface of 128° YX LN substrate, *λ* is the wavelength of the Rayleigh surface wave. Therefore, the wavelength *λ* should be on the order of one or several microns, according to the estimation. Herein, *λ* is desired to be 1.4 μm, then the width of each finger electrode as well as the center distance between each finger electrodes is 1/4 of *λ*, i.e., 350 nm. We adopt the E-beam lithography technique with uniformly coated double-layer E-beam resist (PMMA950K) and conductive adhesive film layer (AR-PC 5090.02) and ultrasonic stripping method to prepare this extremely narrow width IDT. This method can effectively inhibit the collapsing and sticking phenomenon of the pattern prepared by E-beam lithography and improve the quality of UHF SAW devices. The fabricated IDTs are shown in Figure 1b. We chose 20, 40, and 60 pairs of electrodes to emit and receive SAW waves and tested several sensors with IDT apertures of 70, 85, and 100 μm. 

Before the E-beam lithography, it is necessary to decide in advance the number and position of the reflection gratings in the SAW delay line sensor. The distance between the IDT and the reflective grating in the SAW delay line sensor determines the total delay time τ or delay phase ϕ, given by the following equation [18]:(2)$$\tau =\frac{2l}{\nu_{s}}\;\phi =2\pi f\tau ,$$
where *l* is the distance between the IDT and the reflective grating. When a certain environmental parameter changes, the SAW’s propagation velocity along the piezoelectric substrate’s surface changes, thereby inducing a change in delay time in the SAW sensor. Therefore, the parameter to be measured can be determined by monitoring the amount of change in the delay time or phase change. The sensitivity of SAW delay line sensors is positively correlated with the magnitude of τ and can be improved by the flexible design of the reflective grating. The design of a conventional parallel SAW delay line sensor is shown in Figure 1c. We set two reflective gratings parallel to the IDT on the surface, the first short-circuit reflective grating is at a distance of 2010 μm from the IDT, while the second short-circuit reflective grating is at a distance of 2610 μm from the IDT.

In addition, a unique Z-shaped delay line was designed in an attempt to reduce the pitch distance horizontally while also effectively improving the delay time. The acoustic surface wave propagation path in the Z-shaped delay line is shown in Figure 1d. Firstly, the IDT generated a SAW propagating 2010 μm to the first open-circuit reflective grating with an inclination angle of 10.5°, then reflected the SAW at an angle of 21° and transmited 1150 μm to the second inclined open-circuit reflector grating, and then reflects the SAW again, which propagated along a straight line for 600 μm to reach the third short-circuit reflective grating. The reflection angle in the Z-shaped structure was determined based on the angle–phase velocity curve of 128° YX lithium niobate (Figure 1d,e). The reflection angle should not be too small, otherwise the acoustic signals from the IDT and the reflection grids in the aperture direction may interfere with each other. The reflection angle should not be too large either, otherwise the Rayleigh wave phase velocity in the direction of the tilt angle is too small, which may also lead to mutual interference of the reflected incident signals in the time domain.

### 2.2. Numerical Simulation of Electrode Properties

UHF SAW sensors require high uniformity, adhesion, and thickness of the metal film, and the quality of the metal film greatly affects the response of the device [19,20]. In addition, the IDT metallizes the surface of the LN substrate and adds a certain surface mass loading, which changes the actual Rayleigh wave velocity in the substrate [21]. Therefore, we need to determine the electrode material and its thickness and predict the actual operating frequency of the SAW sensor. COMSOL multiphysics is adopted due to its good human–computer interface, strong data visualization, and convenient post-processing [22]. Furthermore, it can easily deal with multi-physical field coupling problems and is suitable for simulating the relevant parameters of SAW sensors based on piezoelectric effect. 

A 2D model composed of 20 pairs of electrodes is developed for preliminary simulations in the frequency domain. Considering the rapid decay of Rayleigh wave energy with depth, the thickness of the model is simplified to 6 μm, which is about four times the wavelength *λ*. Firstly, the electrode material is attempted and set as gold with a thickness of 50 nm; the 2D surface stress and potential distribution obtained are shown in Figure 2a and 2b, respectively. The SAW’s energy distribution was within one to two wavelength depths of the vertical surface, which is in line with the Rayleigh wave characteristics. With the increase of transmission depth, the amplitude of SAW decreases, which verifies the validity of the simplified model.

In the next step, the thickness of the metal electrode was fixed to 50 nm. Then, four metals, Au, Pt, Ag, and Al (material parameters are shown in Table 1), were set as the material for the electrode fingers, respectively, in each simulation. The results are shown in Figure 2c. The center frequency of the SAW sensor decreased sequentially from the order of Al, Ag, Au, and Pt for a finger width 0.35 μm. This phenomenon indicates that under the same structural parameters, the greater the density of the metal electrode material used, the more pronounced the effect of mass is on center frequency. Al has the advantages of low density, low acoustic impedance, and low cost. However, Al electrodes suffer from a low melting point and are highly susceptible to oxidation, resulting in poor robustness. Au has better high-temperature resistance and stability, and its coating technique is more mature than that of Ag and Pt.

After the metal electrode was identified as gold, the effect of different gold film thicknesses on the SAW was explored. Keeping other parameters constant, the simulations were carried out by setting the gold film thickness to 25, 50, 75, and 100 nm, respectively. The effect of different gold film thicknesses on the SAW sensor was obtained, and the results are shown in Figure 2d. It can be seen that for the UHF SAW sensor with a wavelength of 1.4 μm, when the thickness of the metal electrode exceeded 50 nm, the mass loading effect increased significantly, showing a more obvious center frequency shift and admittance became larger. In order to obtain the ultra-high operating frequency, the Rayleigh wave velocity inside the piezoelectric substrate should be attenuated as little as possible by the reflectivity of the electrodes (the variable here is the thickness of the electrodes). Furthermore, their bandwidth was too narrow for the 75 nm and 100 nm thick gold electrodes, compromising the sensor’s response range. However, gold electrodes that are too thin also make the admittance of the sensor too low. As a result, a 50 nm thick gold film was used as the electrode material, and the center frequency of the corresponding model is 2.45 GHz, i.e., the operating frequency of the delay line sensor.

### 2.3. Device Performance and Temperature Test

After the SAW devices are prepared and packaged, they ought to be tested with a network analyzer (Figure 3a) to see if the response matches the design values, such as center frequency, insertion loss, and delay time. Due to the tiny size of the UHF SAW sensors, the network analyzer was used in conjunction with a microwave probe station [24]. As shown in Figure 3b, the microwave probe was fixed on the probe stage through the probe holder and connected to the test port of the vector network analyzer through the coaxial cable. The microwave probe is a chip adapter, completing the conversion from coaxial to microstrip, realizing feeding and detecting input and output signals for the chip, and providing DC bias, adopting a two-contact GS probe head with a contact spacing of 500 μm. When the vector network analyzer was calibrated, we set the center frequency to 2.45 GHz, the scanning frequency bandwidth to 350 MHz, and the number of scanning points to 10,001. Thus, the frequency domain and time domain responses of different delay line sensors were obtained.

The principle of temperature measurement of the SAW delay line sensors is to first determine the delay time or phase of the SAW sensor at the initial temperature, then change the external ambient temperature and continuously record the change in delay time or phase so that the change value corresponds to the temperature value one by one. Since the phase change is more pronounced than the time change, the measurement against the phase change is more accurate. When the phase change passes through the −180° or 180° boundary, it jumps to a new cycle. A special MATLAB program can be used to connect a computer with the vector network analyzer so that continuous phase change data is obtained.

The temperature test platform is shown in Figure 3c, and a sketch of the system is shown in Figure 3d. The sensor was connected to the vector network analyzer using a coaxial cable and then placed in the programmable temperature and humidity test chamber. The test chamber cavity was large, and the heating was uneven, so the thermometer in the test chamber could not accurately feedback the temperature around the SAW sensor. A SHT71 temperature sensor was used in the temperature control next to the SAW sensor as a real-time temperature reference. The SHT71 temperature sensor measures temperature with a precision up to ±0.4 °C and an operating range of −40~120 °C, meeting the basic requirement in the experiment.

## 3. Results and Discussion

### 3.1. Performance Testing 

The results of the preliminary performance tests are presented in two sets of figures. The frequency domain responses for electrode pairs of 20, 40, and 60 are shown in Figure 4a. The frequency domain responses for aperture size of 70, 85, and 100 μm are shown in the other group in Figure 4b. As the number of electrode pairs (or aperture size) increases, the return loss S11 of the SAW sensor tends to decrease, the main bandwidth decreases and the frequency response increases. Therefore, within the scope of this study, we decided to adopt a UHF SAW sensor with 60 electrode pairs and an aperture size of 100 μm for subsequent testing.

Next, we transformed the response in the frequency domain through the discrete Fourier inversion to obtain the response in the time domain using the vector network analyzer. The initial time domain point was set to 0 μs, and the end time domain point was 5 μs. The results of the time domain response of the parallel SAW delay line sensor with 60 pairs of electrodes and an aperture size of 100 μm are shown in Figure 5a. According to the design of the parallel reflective gratings, it can be seen that the distance between the first reflector grating and IDT was 2010 μm, and the estimated delay time was 1.007 μs; the distance between the second reflector grating and IDT was 2610 μm, and the estimated delay time was 1.307 μs. Compared with the vector network analyzer testing results in the time domain, there were obvious excitation signals in the corresponding region, in line with the initial design requirements. The latter two excitation peaks were generated by the secondary reflection of the SAW, which can be eliminated by further design of the reflective gratings.

As mentioned, we also designed a unique Z-shaped delay line sensor. The Z-shaped path is formed by refracting the SAW to a strong short-circuit reflective grating by two inclined open-circuit reflective gratings, and this design can make full use of the piezoelectric substrate. If we adopt a parallel SAW delay line to prepare the same equivalent spacing as the third reflector grating, a distance of 3800 µm in the parallel direction is required. For the Z-shaped delay line, the same sensitivity can be achieved with a delay time of 1.921 µs by maintaining a distance of only 2010 µm in the parallel direction. 

The Z-shaped delay line sensor was also analyzed by placing it in the probe stage and continuing the time domain test using vector network analysis, and an additional reflective grating signal #3 appeared in the 0–5 µs time interval with a peak at 1.921 µs, as is shown in Figure 5b The twice reflections in the Z-shaped delay line result in a signal loss of about −5 to −10 dB, but a good signal peak can still be obtained and the delay time is readable for application in temperature sensing experiments. Furthermore, compared with the traditional parallel delay line, the Z-shaped delay line can reduce the secondary reflection effect generated by the SAW propagation, reduce the occurrence of miscellaneous peaks in the time domain, and improve the anti-interference ability of the sensor.

### 3.2. Temperature Sensitivity Testing

The initial temperature of the test chamber was set to 20 °C, the heating step was 10 °C, the heating time was 15 min, and the final temperature was 80 °C. The real-time test data of the SHT71 temperature sensor is shown in Figure 6a,b. The delay time on the first reflection grating during the initial test was 1.007 μs, and the phase changes in real time are shown in the figure. The phase delay became smaller and smaller when the external ambient temperature rose from 20 °C to 80 °C. This is because the change in temperature also changes the elasticity coefficient and density of the LN substrate, and thermal expansion occurs, which results in the change of the propagation velocity of the SAW on the LN substrate and ultimately affects t the delay phase of the SAW sensor. It can be observed that the phase of the real-time temperature change measured by the SAW sensor was basically the same as that of the SHT71 temperature sensor, indicating that the sensor had good real-time sensing capability of external temperature.

Taking the average of the three results for fitting of the straight line, the results are shown in Figure 7a,b. It can be learned that the phase of the first reflective grating fitted straight line function was equal to *y* = 1361.983 − 63.625*x*, where *y* stands for the temperature *T* and *x* stands for the time *t*. The correlation coefficient was 0.99946, and the temperature sensitivity was 63.625°/°C. The second reflective grating phase fitted straight line function was equal to *y* = 1855.819 − 85.869*x*, the correlation coefficient was 0.99953, and the temperature sensitivity was 85.869°/°C. The greater the absolute value of the slope of the linear fit line, the higher the temperature sensitivity of the sensor and the more pronounced the phase change of the corresponding temperature.

Typically, there is a deviation between the fitted straight line after the linear approximation of the sensor and the actual measured data points. This maximum deviation is the nonlinear error in the sensor, which is often referred to as the relative error and is used as a measure of the linearity of the experimental results, as shown in Equation (3):(3)$$\gamma_{\phi }=\frac{\left|\Delta \phi_{\max}\right|}{y_{FS}}\times 100\%,$$
where $\Delta \phi_{\max}$ is the maximum deviation of the actual experimental results from the fitted straight line and $y_{FS}$ is the maximum change in phase [20]. From the fitted straight line of the first and second reflective gratings, the maximum deviation of the actual measured data from the fitted straight line is found. The temperature sensing nonlinearity of the first reflective grating was only 1.31%, while the temperature test nonlinearity of the second reflective grating was also only 1.21%, which indicates that the SAW delay line sensor had good linearity in temperature sensing.

The hysteresis behavior of a sensor is the difference between the forward and reverse test results for the same environmental variables. The hysteresis behavior of a temperature sensor is measured by Equation (4):(4)$$\gamma_{H}=\frac{\left|\Delta H_{\max}\right|}{y_{FS}}\times 100\%,$$
where $\Delta H_{\max}$ is the maximum difference between the phase at the initial temperature and the phase that recovers to the initial temperature [25]. The temperature hysteresis of the first reflective grating was only 2.17%, while the temperature hysteresis of the second reflective grating was 2.55%, which indicates that the SAW delay line sensor had a good hysteresis behavior and realized reversibility in the temperature sensing.

We also tested the third reflective grating in a Z-shaped delay line with a delay time of 1.921 μs. The same heating process was adopted for the temperature-controlled test chamber, and the results of the fitted curves are shown in Figure 7b. The fitted straight-line function of the third reflective grating was *y* = 2461.350 − 116.685*x*, with a correlation coefficient of 0.99969. The temperature sensitivity was 116.68°/°C, which was higher than that of the traditional parallel delay line sensor. According to Equations (3) and (4), the linearity of the third reflective grating was 1.26%, and the temperature hysteresis was 2.41%, suggesting its outstanding temperature sensing stability as well,

## 4. Conclusions

This paper focuses on the design and fabrication of UHF SAW delay line sensors and their application to real-time temperature detection. The very narrow linewidth IDTs are processed on 128° YX LN piezoelectric substrate by E-beam lithography. We further designed and validated a novel Z-shaped delay line based on the conventional parallel delay line, which successfully improved the sensing sensitivity and reduced the size of the sensor. The temperature sensitivity of the first reflective grating was 63.625°/°C and the temperature sensitivity of the second reflective grating was 85.869°/°C in the operating temperature range of 20 °C to 80 °C. The temperature sensitivity of the three reflective gratings in the Z-shaped delay line was 116.685°/°C. The results prove that the Z-shaped delay line structure can effectively reduce the secondary reflection of the acoustic wave and facilitate the miniaturization and integration of SAW delay line sensors.

## Figures and Tables

**Figure 1 micromachines-15-00135-f001:**
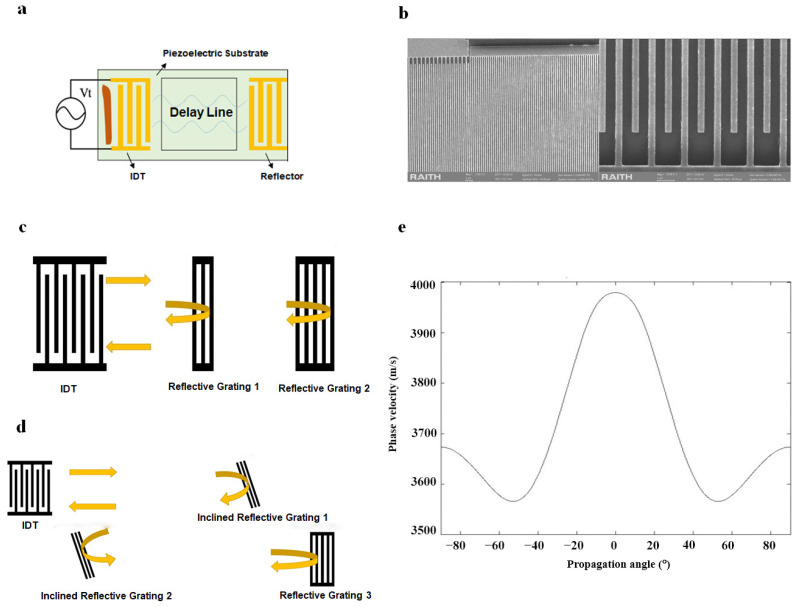
(**a**) Schematic diagram of a SAW delay line device. (**b**) SEM topography images of the fabricated IDTs. (**c**) Parallel SAW delay line sensor. (**d**) Z-shape SAW delay line sensor. (**e**) Phase velocity curve of 128° YX LN substrate at different propagation angles.

**Figure 2 micromachines-15-00135-f002:**
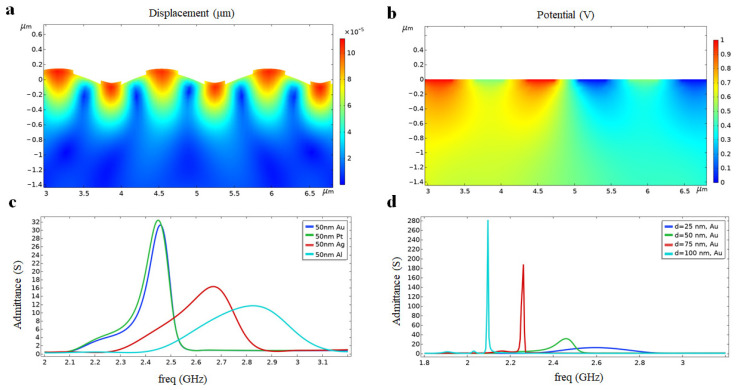
Simulation results of (**a**) the displacement mode and (**b**) the potential distribution of the LN substrate. (**c**) Effect of metallic electrode materials on the admittance curve at a fixed electrode thickness of 50 nm. (**d**) Effect of Au electrode thickness on the admittance curve.

**Figure 3 micromachines-15-00135-f003:**
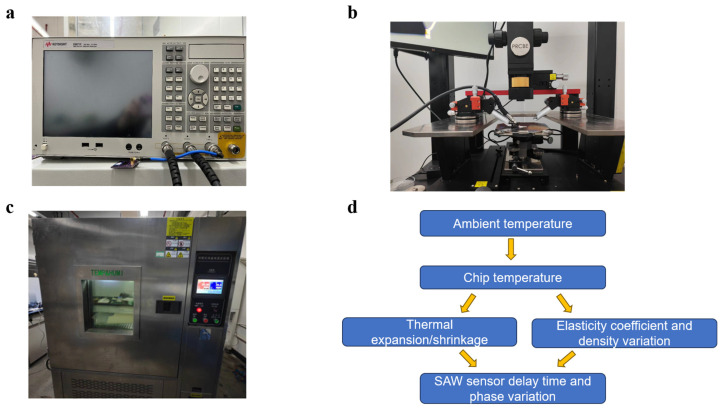
(**a**) E5071C Vector network analyzer. (**b**) Microwave probe station. (**c**) Programmable temperature and humidity-controlled test chamber. (**d**) Temperature sensing procedure.

**Figure 4 micromachines-15-00135-f004:**
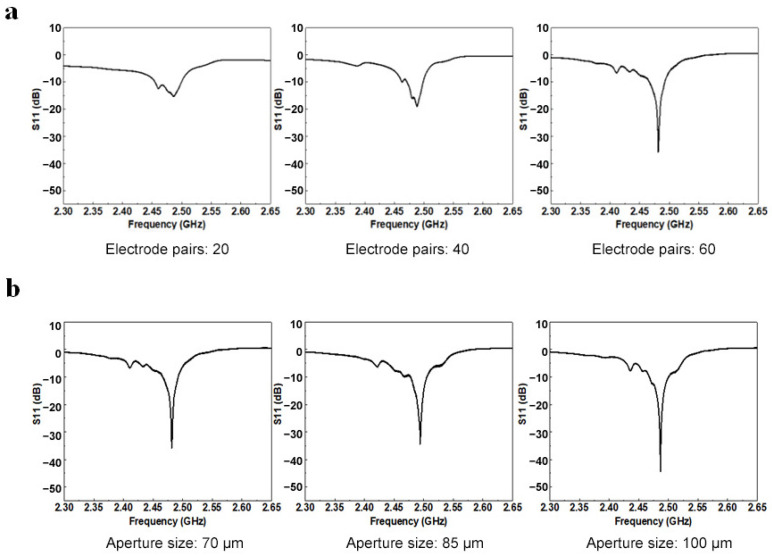
(**a**) Frequency domain response of different numbers of electrode pairs. (**b**) Frequency domain response of different aperture sizes.

**Figure 5 micromachines-15-00135-f005:**
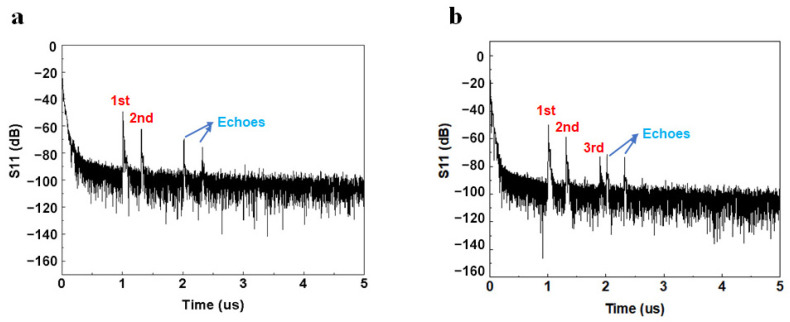
Time-domain signal comparison between (**a**) the parallel delay line and (**b**) the Z-shaped delay line.

**Figure 6 micromachines-15-00135-f006:**
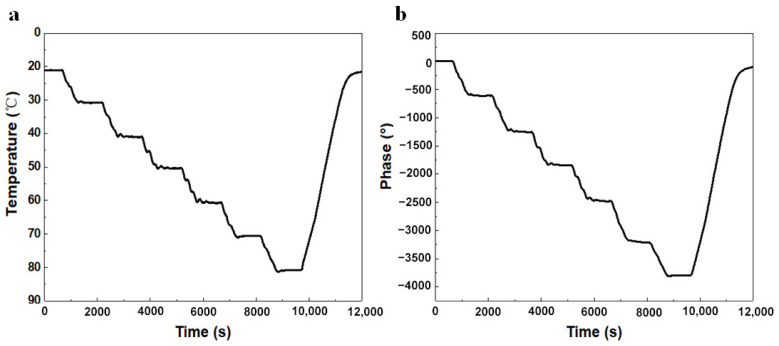
Real-time (**a**) temperature vs. time and (**b**) phase vs. time curves of the sensor.

**Figure 7 micromachines-15-00135-f007:**
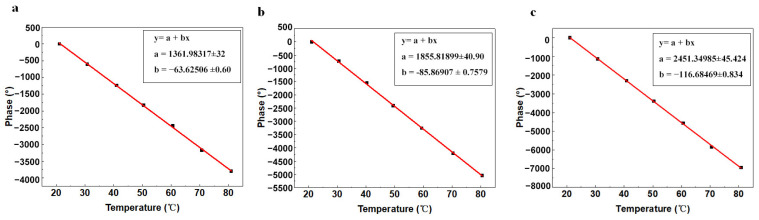
Temperature-phase fitting curves for (**a**) the first parallel reflective grating, (**b**) the second parallel reflective grating, and (**c**) the third Z-shaped reflective grating.

**Table 1 micromachines-15-00135-t001:** Metal–electrode material parameters [23].

Key Parameters	Au	Pt	Ag	Al
Young’s modulus E (GPa)	70	168	83	70.3
Poisson’s ratio μ	0.44	0.38	0.34	0.345
Density r (g/cm^3^)	19.93	21.45	10.5	2.7

## Data Availability

The data that support the findings of this study are available from the corresponding authors upon reasonable request.

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
