# Peer review of "Ultra-High Frequency Surface Acoustic Wave Sensors for Temperature Detection"

_micromachines, 2024, doi:10.3390/mi15010135_

Round 1
Reviewer 1 Report
Comments and Suggestions for Authors
In this manuscript, the authors report an ultrahigh frequency SAW sensor to implement temperature detection. The paper is well-organized and proper simulation and experimental results support these conclusions appropriately. However, aiming at clear and accurate demonstration to the readers, several elements are current missing or need minor revision before the manuscript is accepted.
1. In line 19, the sensitivity value is missed.
2. In line 21, there is some mistakes.
3. In Fig. 1, the legends are too small and not clear; the legend ‘reflecting grating’ has never been used in other parts of the paper, should it be ‘reflective grating’?
4. In line 103 and line 104, you mention the ‘open-circuit grating’ in Fig. 1c). However, in Fig. 1c) you draw a short-circuit grating. It’s confusing.
5. In line 101, the number of magnitude is missed. Check carefully if any other statistics or numbers are missed.
6. In line 15, you introduce that ‘the optimum design parameters of … and reflective gratings are given by numerical simulations.’ However, in line 114, the FEM section, it seems that the design of reflective gratings is not mentioned. Please explain it.
7. Could you give more demonstration on why you choose SAW delay line and not other SAW devices for ultra-high temperature detection in the introduction section?
8. In line 125, the model thickness 6 μm is not four times of the wavelength (1.4 μm).
9. In line 133, the right bracket is missed, and the simulation setup is not clearly expressed. Give more details on mesh size, geometry parameters, study used and so on.
10. The figures in Figure 2 is not clear.
11. In line 163, it should be ‘insertion loss and delay time’.
12. In line 165, What is the model number of the microwave probe you used? And You mentioned that the fabricated delay line sensors are tiny, how tiny are they? Explain them.
13. In this paper, you use the phase delay to determine the sensitivity of the sensor, but how do you determine the phase delay? Give the related equations or explanation for this.
14. In line 218, you put ‘the estimated delay time’. How do you do the calculation?
15. Please give more notations in Fig. 5 for different peaks.
16. In line 150, the ‘um’ should be ‘μm’ instead. Check the spelling of other signals if necessary.
17. From line 204 to line 206, you demonstrate why the sensors with certain number of electrode pairs and apertures are selected for further test, yet the explanation is not clear enough.
Reviewer 2 Report
Comments and Suggestions for Authors
1. What is the main question addressed by the research?
Developing a simple and stable SAW sensor based on ultra-high frequency delay line.
2. Do you consider the topic original or relevant in the field? Does it address a specific gap in the field?
In the paper, the relevant topic in the field of UHF acousto-electronic devices is presented.
3. What does it add to the subject area compared with other published material?
The technology of the extremely short pitch interdigital transducers producing by means of e-lithography.
4. What specific improvements should the authors consider regarding the methodology? What further controls should be considered?
In my opinion, the prospective trend of composite SAW devices using high quality substrate&piezoelectric film transducer obtained by the e-lithography technology could be considered.
5. Are the conclusions consistent with the evidence and arguments presented and do they address the main question posed?
In a frame of the paper's task, the conclusions are consistent with the evidence and arguments presented.
6. Are the references appropriate?
The minimally sufficient list of references reflects a modern results in a subject area.
7. Please include any additional comments on the tables and figures.
In the Table 2, the italics Pt should be corrected.
I hope that Figures 2, 4 and 7 will be presented in the best quality, because the captions and numbers on the axes are poorly readable on them.
Conclusion. I suppose that the experimental results&Comsol modelling obtained in the paper under review become a relevant contribution in the field of UHF SAW acousto-electronic devices.
Reviewer 3 Report
Comments and Suggestions for Authors
This manuscript describes a SAW temperature sensor based on a delay line. The authors uses reflectors to increase the path. Also, in one of the devices, 2 oblique reflectors and one non-oblique reflectors are used, resulting in a larger path. To the knowledge of the reviewer, if reflective delay lines have been used in the past, this configuration of oblique reflector have not been used for sensors. A large work on SAW devices with reflectors and oblique reflectors has been published in the past and at least some of this work should be referenced.
For example Malocha et al, https://www.mdpi.com/1424-8220/13/5/5897/htm demonstrated SAW sensors with identification provided by the reflectors. Plessky et al. (Plessky, Victor, and Marc Lamothe. "Ultra-wide-band SAW RFID/sensors." 2014 European Frequency and Time Forum (EFTF). IEEE, 2014.) describe similar SAW temperature sensors using multiple reflectors. Multiple authors describe SAW temperature sensors using SAW resonators. These papers and maybe other should be referenced.
Regarding the sensor design, the authors chose the electrodes configuration from finite element simulations. The simulation is not really described. Is this a simulation of a transducer, of the full device, a periodic simulation. Is the simulation 2D, 3D? Also, the reason of the choice of the electrode configuration based on the admittance of the transducer is not clearly explained.It is explained p 4 that the metal thickness increases the frequency shift and the conductivity. Which conductivity? Also, why is the frequency shift a problem. This effect is very well know it is simply due to the reflectivity of the electrodes. Figure 2 is of very bad quality and should be improved. Why is the deplacement 2(a) (color) extending outside of some electrodes and why are some electrodes white (not in the colour map). Why is there no electrical potential in the electrode (2b). Are the admittance curves plots of the amplitude of the admittances or plots of the conductance? What is an ideal plot. Also, the characters in this figure are too small to be easily read.
Regarding the design of the oblique reflectors, it is explained that the reflectors are inclined 10.5 degrees resulting in a reflection at 21 degrees. Since lithium niobate is an anisotropic material, the angle of reflection depends on the SAW slowness curve. The angle of reflection is probably different.
Figure 3d shows a schematic of the system (3d). The system includes a pressure sensor but no temperature sensor. This is difficult to understand.
Paragraph 3 describes the choice of the transducer aperture and length. The discussion is difficult to follow. What is meant by "main flap width of the response". The choice looks to be based on the responses of fig. 4 but it is not clear what characterizes a good or a bad response.
Figure 5 show the time domain responses of the sensors. More than one response is shown. The useful response is shown on fig 5b but not on fig 5a. Also, the original of the multiple echos should be discussed.
Paragraph 3.1 discusses the measured results. Several linear functions of the form y=ax+b are given, but the meaning of x and y is not given. Also the text mentions sensitivity of the "third reflective grating delay line". This may be replaced by the "three reflective grating delay line".
In general, it would be interesting to explain why reflective delay lines are used and not resonators. The oblique reflectors device looks new and should be published. Its main advantage is the longer delay for the same size. The discussion in the conclusion mentions "anti-interference ability of the signal". This is not clear.
Comments on the Quality of English Language
see general discussions
